# Comparison of Linear and Convex-Array Transducers in Assessing the Enhancement Characteristics of Suspicious Breast Lesions at Contrast-Enhanced Ultrasound (CEUS)

**DOI:** 10.3390/diagnostics12040798

**Published:** 2022-03-24

**Authors:** Ioana Boca (Bene), Anca Ileana Ciurea, Cristiana Augusta Ciortea, Ștefan Cristian Vesa, Sorin Marian Dudea

**Affiliations:** 1Department of Radiology, “Iuliu Hatieganu” University of Medicine and Pharmacy, 400012 Cluj-Napoca, Romania; ioanaboca90@yahoo.com (I.B.); sdudea1@gmail.com (S.M.D.); 2Department of Radiology, Emergency County Hospital, 400006 Cluj-Napoca, Romania; cristianaciortea@yahoo.com; 3Department of Pharmacology, Toxicology and Clinical Pharmacology, “Iuliu Hatieganu” University of Medicine and Pharmacy, 400012 Cluj-Napoca, Romania; stefanvesa@gmail.com

**Keywords:** contrast-enhanced ultrasound, breast cancer, linear-array transducer, convex-array transducer

## Abstract

The purpose of this study was to determine the observer agreement in assessing the enhancement pattern of suspicious breast lesions with contrast-enhanced ultrasound (CEUS) using high and low frequency transducers. Methods: This prospective study included 70 patients with suspicious breast lesions detected at mammography and/or ultrasound and classified according to the American College of Radiology (ACR) Breast Imaging-Reporting and Data System (BI-RADS) in 4A, 4B, 4C, or 5, who underwent CEUS examinations between October 2020 and August 2021. Results: Participants’ ages ranged from 28 to 83 years (48.5 + 6.36, mean age + SD). We obtained a substantial agreement for the first reader (kappa = 0.614, *p* < 0.001) and a perfect agreement for the second and third reader (kappa = 1, *p* < 0.001) between the two transducers for the uptake pattern. A moderate agreement for the second and third reader (kappa = 0.517 and 0.538, respectively, *p* < 0.001) and only a fair agreement (kappa = 0.320, *p* < 0.001) in the case of the first reader for the perilesional enhancement was observed. We obtained an excellent inter-observer agreement (Cronbach’s Alpha coefficient = 0.960, *p* < 0.001) for the degree of enhancement, a good inter-observer agreement for the uptake pattern and perilesional enhancement (Cronbach’s Alpha coefficient = 0.831 and 0.853, respectively, *p* < 0.001), and a good and acceptable inter-observer agreement for internal homogeneity, perfusion defects and margins of the lesions (Cronbach’s Alpha coefficient = 0.703, 0.703 and 0.792, respectively, *p* < 0.001) concerning the evaluation of breast lesions with the linear-array transducer. Conclusions: The evaluation of suspicious breast lesions by three experts with high-frequency linear-array transducer and low-frequency convex-array transducer was comparable in terms of uptake pattern and perilesional enhancement. The agreement regarding the evaluation of the degree of enhancement, the internal homogeneity, and the perfusion defects varied between fair and substantial. For all CEUS characteristics, the inter-observer agreement was superior for linear-array transducer, which leads to more a homogeneous and reproducible interpretation.

## 1. Introduction

Contrast-enhanced ultrasound (CEUS) is an imaging technique with many applications in the clinical practice, considered safe for intravenous administration in adults and children, and also for intracavitary use. Initially, it was implemented and developed for hepatic imaging, with the aim of differentiating benign and malignant liver lesions, and for detecting liver metastases [1].

Among other clinical applications of CEUS, breast pathology is an active area of research, but the use of CEUS in clinical practice is not yet recommended [2]. 

There are studies in the literature that have demonstrated the usefulness of CEUS in differentiating benign and malignant breast lesions, even found correlations with pathological characteristics (estrogen and progesterone receptors, histological grade, human epidermal growth factor receptor 2, Ki-67) [3].

Sonovue^®^ has a bubble size distribution between 0.7–10 μm in diameter. Gorge et al. [4] found that microbubbles smaller than 2 μm represent less than 5% of the global gas volume, and their effects were negligible at frequencies up to 7 MHz. Furthermore, 80% of the echogenicity was provided by bubbles with 3–9 μm diameter, which proved to be well adapted to clinical applications in the usual medical frequency range (1–7 MHz). 

Considering the fact that it was observed that contrast microbubbles resonate better at frequencies lower that 10 MHz [5], this study aims to compare the agreement between different types of transducers with high frequency (frequency range of 3–8 MHz) and low frequency (frequency range of 1–5 MHz) regarding the enhancement pattern at CEUS of suspicious breast lesions.

To the best of our knowledge, there is only a study published in the literature regarding the assessment of breast lesions at CEUS by using the linear-array and convex-array transducer. Starting from the idea that the convex-array transducer, having lower frequencies could be more suitable in the evaluation of breast lesions, the originality of the present study is represented by the comparison of observer agreement regarding the evaluation of suspicious breast lesions by using the two types of transducers. 

The clinical benefit of this study would be to adapt the CEUS examination protocol in order to obtain reliable and reproducible results.

## 2. Materials and Methods

This prospective study was approved by the institutional review board (Number/Date: 280/11 August 2020) and informed consent was obtained from each patient before performing CEUS.

### 2.1. Study Design and Population

Between October 2020 and August 2021, 84 patients underwent CEUS examinations. 

We included only patients with suspicious breast lesions detected at mammography and/or ultrasound and classified according to the American College of Radiology (ACR) Breast Imaging-Reporting and Data System (BI-RADS) in 4A, 4B, 4C, or 5. All included patients agreed to perform the biopsy and did not have contraindications for the administration of contrast media such as hypersensitivity to sulfur hexafluoride (or any of the components of SonoVue^®^), acute cardiac failure, recent acute coronary syndrome or clinically unstable ischemic cardiac disease, known right-to-left shunts, severe rhythm disorders or pulmonary hypertension (pulmonary artery pressure > 90 mmHg), uncontrolled systemic hypertension and respiratory distress syndrome. 

For all biopsied lesions, we had the pathology reports, such as the case of malignant lesions included the histological tumor grade, the estrogen receptor (ER), progesterone receptor (PR), and human epidermal growth factor receptor 2 (HER2) status and the value of Ki-67.

We excluded from the study patients without contrast agent in the lesion or surrounding parenchyma later found with brachial vein thrombosis, patients who did not perform both CEUS examinations using linear-array and convex-array transducer and lesions under 11 mm diameter due to the difficulty of visualizing them with the convex-array transducer.

Figure 1 summarizes the flowchart of patient’s selection.

### 2.2. Image Acquisition

All breast CEUS examinations were acquired with a LOGIQ S8 ultrasound machine (General Electric Ultrasound Korea, Seongnam-si, Korea) using a low-frequency convex-array transducer (probe C1-5) with a frequency range of 1–5 MHz and a high-frequency linear array transducer (probe 9L-D) with a frequency range of 3–8 MHz.

Before the contrast examination, the lesion was assessed on both gray-scale and color Doppler ultrasound using a ML6-15 probe (frequency range of 4–15 MHz) to identify the scanning plane with the largest diameter of the lesion and with the richest vascularization. Areas with acoustic attenuation and areas with ultrasound visible microcalcifications were avoided. 

We performed two contrast-enhanced ultrasounds on each patient in the same session. We started evaluating the lesions with the 9L-D transducer and after 10 min of waiting, we moved to the second examination using the C1-5 transducer. While using the contrast software an exact frequency could not be selected. To adapt the frequencies there were only the options “resolution”, “general” or “penetration”, therefore in order to avoid overlapping frequencies between the two transducers, we used “resolution” for 9L-D probe and “penetration” for C1-5 probe.

In the case of both examinations, we set the mechanical index (MI) at 0.06, the gain of 100–120 dB, single focus, image depth of approximately 3–4 cm, and SonoVue^®^ (Bracco, Amsterdam, The Netherlands) was administered intravenously as a bolus of 2.4 mL followed by a flush of 5 mL sodium chloride 0.9% using a 20 gauge cannula. 

To avoid artifacts, before the examination we asked the patients not to speak or move and avoid coughing or extensive breathing movements during the image acquisition. 

All CEUS examinations were performed by the same physician, before the biopsy and the images were recorded for at least 180 s.

### 2.3. Image Interpretation

All recorded examinations were independently evaluated by three radiologists, one that performed the examinations (third reader) and the other two who reviewed only the images, without having any information regarding the patients. 

The lesions were assessed in terms of enhancement degree compared with the surrounding breast tissue (non-, hypo-, iso-, hyper-enhancement), internal homogeneity (homogeneous/inhomogeneous), presence/absence of perfusion defects, uptake pattern (centripetal/centrifugal), lesion margins (circumscribed/non-circumscribed) and presence/absence of perilesional enhancement. 

### 2.4. Statistical Analysis

Statistical analysis was performed using the MedCalc^®^ Statistical Software version 20.011 (MedCalc Software Ltd., Ostend, Belgium; https://www.medcalc.org; accessed on 10 February 2022). Nominal variables were characterized by frequency and percentage. Cohen’s kappa coefficient was used in order to assess the agreement between two transducers. For inter-rater agreement between examiners, we used intraclass correlation coefficient, for more than two examiners. A *p*-value < 0.05 was considered statistically significant.

All qualitative variables were coded as follows: enhancement degree (non-enhancement = 0, hypo-enhancement = 1, iso-enhancement = 2, hyper-enhancement = 3), internal homogeneity (homogeneous = 0, inhomogeneous = 1), presence of perfusion defects = 1, absence of perfusion defects = 0, uptake pattern (centripetal = 1, centrifugal = 0), lesion margins (circumscribed = 0, non-circumscribed = 1) and presence of perilesional enhancement = 1, absence of perilesional enhancement = 0. 

## 3. Results

A total of 70 patients with suspicious breast lesions detected at mammography and/or ultrasound were included in the study. Participants’ ages ranged from 28 to 83 years (48.5 + 6.36, mean age + SD). All the characteristics regarding the patients are listed in Table 1.

We evaluated the agreement between the linear-array transducer and the convex-array transducer regarding the lesion’s enhancement pattern for each of the three readers. The results are listed in Table 2.

We could not obtain a kappa value for the margins of the lesions in the case of first reader since at the convex-array transducer there were no lesions characterized as having circumscribed margins.

We evaluated the inter-observer agreement between the three readers related to the linear-array transducer and the convex-array transducer (Table 3).

## 4. Discussion

Our results showed that most of the lesions evaluated by all three readers with both high frequency and low frequency transducers presented hyper-enhancement, inhomogeneous aspect, non-circumscribed margins, perfusion defects, centripetal uptake pattern, and perilesional enhancement.

These findings were consistent with literature data considering the evaluation of the lesions through linear-array transducer. In particular, most of the lesions included in the study (88.57%) were found to be malignant after biopsy [3,6,7,8,9]. 

Vraka et al. [3] found that 96.3% of lesions had a inhomogeneous enhancement, 64.5% had ill-defined margins, 76.5% had perilesional enhancement and all the lesions had perfusion defects and a centripetal uptake pattern when assessing them with the linear-array transducer. In our study at linear-array and convex-array transducer, according to the first, second and third reader, 98.1%, 100% and 100% of lesions were inhomogeneous, 95.5%, 96.7% and 100% had ill-defined margins, 81%, 75% and 85.2% had perilesional enhancement, 98.1%, 100% and 100% had perfusion defects, and 95.7%, 100% and 100% of the lesions had a centripetal uptake pattern.

A single lesion was described by all examiners as hyper-enhanced when evaluated with the linear-array transducer and non-enhanced at the convex-array transducer. We could not find an explanation in this case, neither from a technical point of view because the examination protocol was respected and there were no problems with the administration of the contrast agent, nor from a pathological point of view (luminal B subtype breast cancer).

Although the inhomogeneous appearance at CEUS was more suggestive for malignant lesions, we also found breast cancers that presented internal homogeneity. In our study, one lesion was considered by the three readers to be homogeneous when evaluated with both transducers. It was the case of a young patient, who had a large breast mass, with rich vascularization, luminal B subtype, positive ER, negative PR, Ki-67 of 40%, tumor cellularity of 80%, and no areas of necrosis. The homogeneous aspect, in this case, could be explained by the intense vascularization, hypercellularity in the lesion, and the status of positive estrogen receptors. Breast cancers with positive estrogen receptors have a lower rate of cell proliferation, so less necrotic tissue, which also explains the homogeneous appearance at CEUS. On the other hand, breast cancers with negative ER present areas of central necrosis and fibrosis, therefore an inhomogeneous appearance at CEUS [6,7,10,11,12].

The three readers classified a malignant lesion as inhomogeneous with perfusion defects at the linear-array transducer and homogeneous without any perfusion defects at the convex-array transducer. It was the case of a luminal A subtype, with positive ER and PR, Her2 negative, and Ki-67 < 20%, that according to the literature should have a homogeneous appearance when evaluated with the linear-array transducer [10,13]. The inhomogeneous appearance of lesions when evaluated with the linear-array transducer and the homogeneous appearance when evaluated with the convex-array transducer could be explained by the fact that microbubbles were made to resonate better at lower frequencies or the lower resolution of convex-array transducer gives the impression of homogeneous filling of the lesions [4,5]. 

The same explanation could be taken into account regarding the perilesional enhancement. In most cases included in the study, the perilesional enhancement was observed with both transducers. However, regarding the discrepancies in the evaluation of the perilesional enhancement between the two transducers, all three readers observed the presence of the perilesional enhancement at the convex-array transducer and the absence of perilesional enhancement at the linear-array transducer. A recent study conducted by Piskunowicz et al. [14] found that in most cases, the default CEUS MI setting by the manufacturer is often suboptimal, and by increasing the acoustic power, the MI, the microbubble signal intensity and image quality improved. The amount of contrast media we administered was 2.4 mL, according to studies found in the literature [7,15,16,17,18], but a recent meta-analysis showed that diagnosis of CEUS examinations may be more accurate if >3 mL of SonoVue^®^ is injected [19]. Therefore, it is possible that by using a higher dose of contrast medium, the results may be different.

In our study, three breast lesions were classified by all examiners as having a centrifugal pattern at both transducers, two were fibroadenomas and one a DCIS + borderline phyllodes. We also found two fibroadenomas without enhancement and two hyper-enhanced fibroadenomas with a centripetal uptake. Furthermore, the radial scar included in our study is an example of a false positive result on gray-scale ultrasound appearing as a hypoechoic, spiculiform lesion, associated with architectural distortion, but without enhancement at CEUS. These aspects were consistent with the literature, according to which at the linear-array tranducer benign lesions are non-enhancing or have a centrifugal uptake pattern [20,21,22,23,24,25]. In addition, Li et al. [15] and Luo et al. [13] used only the linear-array transducer and observed cases of fibroadenomas with hyper-enhancement in their studies, which could lead to false-positive results in CEUS. 

Regarding the agreement between the two transducers, the uptake pattern was the one that correlated best, with a substantial agreement for the first reader and a perfect agreement for the second and third reader. The perilesional enhancement evaluated with the two transducers had a moderate agreement for the second and third reader and only a fair agreement in the case of the first reader. Regarding the degree of enhancement, the internal homogeneity, and the perfusion defects, a variable agreement was obtained between the two transducers, from a fair agreement for the first reader to a moderate agreement for the second reader and substantial agreement for the third reader. To the best of our knowledge, there is only one study in the literature that compared the effectiveness of linear and convex transducers in assessing breast lesions [26]. They did not notice a significant difference between the two transducers regarding benign lesions, but in the case of malignant lesions, the convex-array transducer detected better the perfusion defects and the surrounding vessels. One aspect that needs to be mentioned is that as a linear-array transducer they used the probe L12-5 with a frequency range of 5–12 MHz, while we used in our study the probe 9L-D with a frequency range of 3–8 MHz. We initially tried using probe ML6-15, but we did not notice any contrast microbubbles in the lesion or the surrounding tissue, although the parameters (mechanical index, gain, focus, depth) were set accordingly, therefore we assumed that the high frequency shattered the microbubbles.

Concerning the inter-observer agreement between the three readers concerning the evaluation of suspicious breast lesions with the linear-array transducer, an excellent inter-observer agreement was obtained for the degree of enhancement, a good inter-observer agreement for the uptake pattern and perilesional enhancement, and a good and acceptable inter-observer agreement for internal homogeneity, perfusion defects and margins of the lesions. With the convex-array transducer, a good inter-observer agreement was observed for the degree of enhancement and uptake pattern, a good and acceptable inter-observer agreement for internal homogeneity, perfusion defects, and perilesional enhancement. Related to the margins of the lesions, a not acceptable inter-observer agreement was obtained for the convex-array transducer, which could be explained by the fact that due to the lower frequency and lower resolution, it is more difficult to appreciate them, especially in the case of small breast lesions. Therefore, it is preferable to use the linear-array transducer in evaluating breast lesions because the interpretation is more homogeneous and reproducible. Radiomics is a new field that has proven to be useful in radiology, even when applied to different imaging techniques, to distinguish between benign and malignant lesions or even between different histopathological types of cancer by using artificial intelligence [27,28,29,30,31]. Thus, an aspect to be considered in the following studies could be represented by the use of radiomics in the evaluation of CEUS images in order to predict the pathological characteristics of the lesions.

We acknowledge that our study has some limitations. It is a single institution study with small sample size and we also included suspicious lesions that proved to be benign after biopsy.

## 5. Conclusions

The evaluation of suspicious breast lesions by three experts with high-frequency linear-array transducer and low-frequency convex-array transducer was comparable in terms of uptake pattern and perilesional enhancement. The agreement regarding the evaluation of the degree of enhancement, the internal homogeneity, and the perfusion defects varied between fair and substantial. For all CEUS characteristics, the inter-observer agreement was superior for linear-array transducer, which leads to more a homogeneous and reproducible interpretation.

## Figures and Tables

**Figure 1 diagnostics-12-00798-f001:**
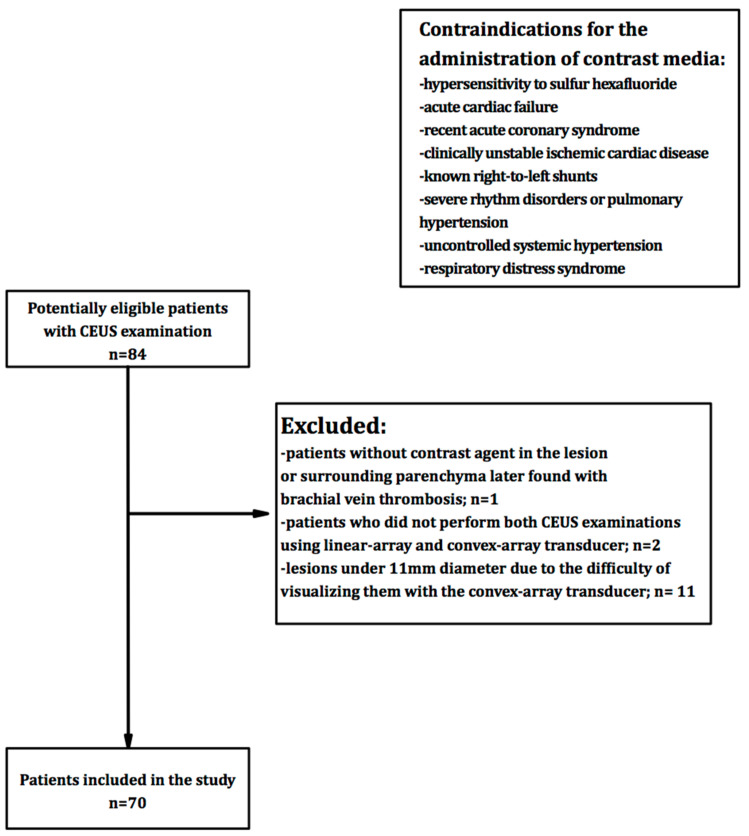
Flowchart of patient’s inclusion in the study.

**Table 1 diagnostics-12-00798-t001:** Patient’s characteristics.

Patient’s Characteristics	Number	Frequency (%)
**Mean age, years (range)**	48.5 (28–83)	
**Mean lesion size, mm (range)**	26.37 (11–87.5)	
**Pathological Type**		
**Benign**	8	**11.43**
Fibroadenoma	6	8.57
Radial scar	1	1.43
Intraductal papilloma	1	1.43
**Malignant**	62	**88.57**
DCIS	1	1.43
DCIS + Borderline phyllodes	1	1.43
IDC NST	53	75.72
IDC with mucinous components	1	1.43
Tubular carcinoma	1	1.43
Papillary carcinoma	2	2.85
Malignant phyllodes	1	1.43
ILC	2	2.85
**Nottingham grade**		
I	8	13.12
II	34	55.73
III	19	31.15
**Estrogen receptor status**		
Positive	49	80.32
Negative	12	16.68
**Progesterone receptor status**		
Positive	29	47.54
Negative	32	52.46
**Her2 status**		
Positive	35	57.38
Negative	26	42.62
**Ki-67**		
<20%	26	42.62
≥20%	35	57.38
**Luminal subtype**		
A	22	36.07
B	27	44.27
Her2+	6	9.83
Triple-negative	6	9.83

**Table 2 diagnostics-12-00798-t002:** Agreement between the linear-array transducer and convex-array transducer regarding lesion’s enhancement pattern for the first, second and third reader.

Lesion’s Enhancement Pattern at CEUS	First Reader		Second Reader		Third Reader	
	Kappa Coefficient	*p*-Value	Kappa Coefficient	*p*-Value	Kappa Coefficient	*p*-Value
Enhancement degree (hypo-, iso-, hyper-enhancement)	0.320	<0.001	0.644	<0.001	0.498	<0.001
Internal homogeneity (homogeneous/inhomogeneous)	0.301	0.015	0.711	<0.001	0.485	<0.001
Presence/absence of perfusion defects	0.300	0.017	0.711	<0.001	0.485	<0.001
Uptake pattern (centripetal/centrifugal)	0.614	<0.001	1	<0.001	1	<0.001
Lesion margins (circumscribed/non-circumscribed)	-	-	0.841	<0.001	1	<0.001
Presence/absence of perilesional enhancement	0.320	0.002	0.517	<0.001	0.538	<0.001

Kappa < 0—no agreement; Kappa 0–0.2—slight agreement; Kappa = 0.21–0.40—fair agreement; Kappa = 0.41–0.60—moderate agreement; Kappa = 0.61–0.80—substantial agreement; Kappa = 0.81–1—almost perfect agreement.

**Table 3 diagnostics-12-00798-t003:** Inter-observer agreement between the three readers related to the linear-array and convex-array transducer.

Lesion’s Enhancement Pattern at CEUS	Linear-Array Transducer		Convex-Array Transducer	
	Cronbach’s Alpha Coefficient	*p*-Value	Cronbach’s Alpha Coefficient	*p*-Value
Enhancement degree (hypo-, iso-, hyper-enhancement)	0.960	<0.001	0.903	<0.001
Internal homogeneity (homogeneous/inhomogeneous)	0.703	<0.001	0.757	<0.001
Presence/absence of perfusion defects	0.703	<0.001	0.757	<0.001
Uptake pattern (centripetal/centrifugal)	0.831	<0.001	0.832	<0.001
Lesion margins (circumscribed/non-circumscribed)	0.792	<0.001	0.330	<0.001
Presence/absence of perilesional enhancement	0.853	<0.001	0.762	<0.001

Cronbach’s Alpha = 0.01–0.60—not acceptable internal consistency; Cronbach’s Alpha = 0.61–0.70—acceptable internal consistency; Cronbach’s Alpha = 0.71–0.80—good and acceptable internal consistency; Cronbach’s Alpha = 0.81–0.90—good internal consistency; Cronbach’s Alpha = 0.91–1—excellent internal consistency.

## Data Availability

The data is available only by request.

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
