# Peer review of "Comparison of Linear and Convex-Array Transducers in Assessing the Enhancement Characteristics of Suspicious Breast Lesions at Contrast-Enhanced Ultrasound (CEUS)"

_diagnostics, 2022, doi:10.3390/diagnostics12040798_

Round 1

Reviewer 1 Report

The purpose of this study is to compare the agreement between low and high frequency transducers in imaging various breast lesions with CEUS with SonoVue. The agreement on 6 parameters was evaluated for 3 observers. The clinical benefits of the study are unclear and the issue (scientific or clinical) that is supposedly addressed is not presented at all. The study lacks focus and is poorly designed.

Major issues:

  1. Improve introduction. Need more information on the Sonovue and the contrast microbubbles. Three lines are not sufficient for the reader who is not fully familiar with the topic (breast imaging by CEUS with Sonovue) to understand the problems/limitations associated with this method, the issues that remain to be solved before it becomes clinical practice, the issues that are addressed by this study and the significance of the study.
  2. A meta-analysis study of CEUS with Sonovue for breast lesions (Lu et al, Technol Cancer Res Treat, 2020) showed that "The dose of contrast agent used in CEUS examinations could affect diagnostic performance. Diagnosis of CEUS examinations may be more accurate if > 3 ml of SonoVue is bolus-injected once within the recommended dose range." How is the 2.4 ml dose justified in this study?
  3. Was there an overlap in operating frequencies between the two transducers? Namely, one operated at 1-5 MHz and the other at 3-8 MHz. Did you select non-overlapping frequencies and which ones? If the aim of the study is to compare operating frequencies, then at what frequencies were the 6 parameters evaluated? For example, the results in Tables 2-6 are at what frequencies?
  4. More important is the question that arises from the statement in line 55 where "Sonovue ... resonates better at about 4 MHz". If this is the case, why bother looking at lower or higher frequencies? No justification is provided. In addition, the reference included for this (#4) shows results for 12-43 MHz and not 4 MHz, which is very confusing.
  5. Is it certain that results are due to frequency differences and not type of transducer?
  6. It is unclear how kappa and cronbach's alpha were estimated. How did you represent the qualitative variables for the calculations? Was, for example, homogenous set as 1 and inhomogeneous as 0? Should be explained.
  7. It is expected that the first reader, who performed the examination, will have some bias in his/her readings relative to the other two. So, are they comparable? Tables 2-4 actually show that reader 1 differs significantly from the other two in his/her assessment of the images from the two transducers and it is surprising to see the good interobserver agreement in Tables 5 and 6. How can this be explained?

Minor issues:

  1. I recommend moderated editing of English because the sentences are not well structured and are often confusing. For examples, lines 43-45, do you mean that CEUS: "It was initially developed and implemented for hepatic imaging and specifically for the detection of liver metastases and the differentiation between benign and malignant hepatic lesions. "? Or was it developed for other applications as well but was mostly successful in hepatic imaging?
  2. lines 46-47 ... wrong use of "therefore", the whole sentence should be rewritten.
  3. Line 95: What are the "macrocalcifications"? What size do you imply by macro? I assume you mean something else and not "microcalcifications" since you have DCIS cases.
  4. "Inter-observer agreement" is a better scientific term than "internal consistency among observers".
  5. Line 261: "... the agreement ... was variable among multiple readers." is very poor and non-scientific statement.
  6. How is radiomics involved in this imaging technique and how does it relate to the pathological characteristics of the lesions? (lines 251-253). Five references are listed for this, one being in US, and the other 4 in other modalities. What is the relevance? 
  7. Use past-tense throughout your paper or, at a minimum, be consistent.

Author Response

The purpose of this study is to compare the agreement between low and high frequency transducers in imaging various breast lesions with CEUS with SonoVue. The agreement on 6 parameters was evaluated for 3 observers. The clinical benefits of the study are unclear and the issue (scientific or clinical) that is supposedly addressed is not presented at all. The study lacks focus and is poorly designed.

Major issues:

  1. POINT 1: Improve introduction. Need more information on the Sonovue and the contrast microbubbles. Three lines are not sufficient for the reader who is not fully familiar with the topic (breast imaging by CEUS with Sonovue) to understand the problems/limitations associated with this method, the issues that remain to be solved before it becomes clinical practice, the issues that are addressed by this study and the significance of the study.

Response Point 1: Thank you for the suggestions. As a response to Point 1, we added in the body text the following statements:

“Sonovue®has a bubble size distribution between 0.7–10 μmin diameter. Gorge et al found that microbubbles smaller than 2 μmrepresent less than 5% of the global gas volume, and their effects was negligible at frequencies up to 7 MHz. Furthermore, 80% of the echogenicity was provided by bubbles with 3-9 μmdiameter, which proved to be well adapted to clinical applications in the usual medical frequency range (1-7MHz).” (lines 77-82)

“To the best of our knowledge, the studies published in the literature regarding the assessment of breast lesions at CEUS consist in the use of linear-array transducer. Starting from the idea that the convex-array transducer, having lower frequencies could be more suitable in the evaluation of breast lesions, the originality of the present study is represented by the comparison between the two types of transducers regarding the evaluation of suspicious breast lesions.

The clinical benefit of this study would be to adapt the CEUS examination protocol of in order to obtain reliable and reproducible results.” (lines  88-95)

  1. POINT 2: A meta-analysis study of CEUS with Sonovue for breast lesions (Lu et al, Technol Cancer Res Treat, 2020) showed that "The dose of contrast agent used in CEUS examinations could affect diagnostic performance. Diagnosis of CEUS examinations may be more accurate if > 3 ml of SonoVue is bolus-injected once within the recommended dose range." How is the 2.4 ml dose justified in this study?

Response Point 2: Thank you for the suggestions. As a response to Point 2, we added in the body text the following statements: “The amount of contrast media we administered was 2.4 ml, according to studies found in the literature (7,15–18)but a recent meta-analysis showed that diagnosis of CEUS examinations may be more accurate if >3 ml of SonoVue®is injected(19). Therefore it is possible that using by a higher dose of contrast medium the results may be different.” (lines 380-384)

  1. POINT 3: Was there an overlap in operating frequencies between the two transducers? Namely, one operated at 1-5 MHz and the other at 3-8 MHz. Did you select non-overlapping frequencies and which ones? If the aim of the study is to compare operating frequencies, then at what frequencies were the 6 parameters evaluated? For example, the results in Tables 2-6 are at what frequencies?

Response Point 3: Thank you for the suggestions. As a response to Point 3, we took into account this aspect and we added in the body text the explanation contained in the following statement:“While using the contrast software an exact frequency could not be selected. To adapt the frequencies there were only the options "resolution", "general" or "penetration". To avoid overlapping frequencies between the two transducers, we used "resolution" for 9L-D probe and "penetration" for C1-5 probe.” (lines 163-170)

  1. POINT 4: More important is the question that arises from the statement in line 55 where "Sonovue ... resonates better at about 4 MHz". If this is the case, why bother looking at lower or higher frequencies? No justification is provided. In addition, the reference included for this (#4) shows results for 12-43 MHz and not 4 MHz, which is very confusing.

Response Point 4: Thank you for the observation. As a response to Point 4, in the cited article Figure 1 illustrates the resonance frequency of SonoVue according to the diameter of microbubbles and reveals that SonoVue has a resonance frequency below 10 MHz (better at approx. 4 MHz) .

  1. POINT 5: Is it certain that results are due to frequency differences and not type of transducer?

Response Point 5: Considering the fact that we selected "resolution" for 9L-D probe and "penetration" for C1-5 probe (as mentioned in PONIT 3), we can say that the results are due to frequency differences and not type of transducer even without knowing the exact frequency.

  1. POINT 6: It is unclear how kappa and cronbach's alpha were estimated. How did you represent the qualitative variables for the calculations? Was, for example, homogenous set as 1 and inhomogeneous as 0? Should be explained.

Response Point 6: Thank you for the suggestions. As a response to Point 6, we added in the body text the following statement:“All qualitative variables were coded as follows: enhancement degree (non- enhancement =0, hypo-enhancement =1, iso-enhancement=2, hyper-enhancement=3), internal homogeneity (homogeneous=0,inhomogeneous=1), presence of perfusion defects=1, absence of perfusion defects=0, uptake pattern (centripetal=1,centrifugal=0), lesion margins (circumscribed=0, non-circumscribed=1) and presence of perilesional enhancement=1, absence of perilesional enhancement=0.” (lines 197-203)

  1. POINT 7: It is expected that the first reader, who performed the examination, will have some bias in his/her readings relative to the other two. So, are they comparable? Tables 2-4 actually show that reader 1 differs significantly from the other two in his/her assessment of the images from the two transducers and it is surprising to see the good interobserver agreement in Tables 5 and 6. How can this be explained?

Response Point 7: Thank you for the suggestions. As a response to Point 7, we mentioned in the body text that the third reader is the one that performed the examination (line 183). It is true that the first reader had a weaker agreement between the transducers. The other two readers had a slightly better agreement between the transducers, but this was moderate in most situations. As for the agreement on each type of transducer between examiners, it was from the fair upwards. The agreement for convex transducer was lower between readers, for margin lesions and perilesional enhancement. Examination with the linear transducer is more likely to provide more easily reproducible results for examiners.

Minor issues:

  1. PONT 1: I recommend moderated editing of English because the sentences are not well structured and are often confusing. For examples, lines 43-45, do you mean that CEUS: "It was initially developed and implemented for hepatic imaging and specifically for the detection of liver metastases and the differentiation between benign and malignant hepatic lesions. "? Or was it developed for other applications as well but was mostly successful in hepatic imaging?

Response Point 1: Thank you for the suggestions. As a response to Point 1, we modified in the body text the following statements: “Initially, it was implemented and developed for hepatic imaging, with the aim of differentiating benign and malignant liver lesions, and for detecting liver metastases” (lines 44-71)

  1. POINT 2: lines 46-47 ... wrong use of "therefore", the whole sentence should be rewritten.

Response Point 2: Thank you for the suggestions. As a response to Point 2, we modified in the body text the following statements: “Among other clinical applications of CEUS, breast pathology is an active area of research, but the use of CEUS in clinical practice is not yet recommended.” (lines 72-73)

  1. POINT 3: Line 95: What are the "macrocalcifications"? What size do you imply by macro? I assume you mean something else and not "microcalcifications" since you have DCIS cases

Response Point 3: Thank you for the suggestions. As a response to Point 3, we modified in the body text the following statements: “Areas with acoustic attenuation and areas with ultrasound visible microcalcifications were avoided.” (line 159)

  1. POINT 4: "Inter-observer agreement" is a better scientific term than "internal consistency among observers".

Response Point 4: Thank you for the suggestions. As a response to Point 4 we modified the body text "internal consistency among observers" with "Inter-observer agreement".

  1. POINT 5: Line 261: "... the agreement ... was variable among multiple readers." is very poor and non-scientific statement.

Response Point 5: Thank you for the suggestions. As a response to Point 5 we modified the conclusion text the following statements“The evaluation of suspicious breast lesions with high-frequency linear-array transducer and low-frequency convex-array transducer was comparable in terms of uptake pattern and perilesional enhancement. The agreement regarding the evaluation of the degree of enhancement, the internal homogeneity, and the perfusion defects varied between fair and substantial. For all CEUS characteristics, the inter-observer agreement between all readers was superior for linear-array transducer, which leads to more a homogeneous and reproducible interpretation.” (lines 464-470)

  1. POINT 6: How is radiomics involved in this imaging technique and how does it relate to the pathological characteristics of the lesions? (lines 251-253). Five references are listed for this, one being in US, and the other 4 in other modalities. What is the relevance? 

Response Point 6: Thank you for the suggestions. As a response to Point 6 we modified we modified in the body text the following statements:“Radiomics is a new field that has proven to be useful in radiology, even when applied to different imaging techniques, to distinguish between benign and malignant lesions or even between different histopathological types of cancer by using artificial intelligence.” (lines 454-457)

The idea was that if radiomics proved to be successfully used in all imaging techniques, it could be valuable applied to CEUS images.

  1. POINT 7: Use past-tense throughout your paper or, at a minimum, be consistent.

Response Point 7: Thank you for the suggestions. As a response to Point 7 we modified we modified in the body text the by using past-tense as much as it was possible.

Reviewer 2 Report

Manuscript ID: Diagnostics - 1616884

Title:  Comparison of linear and convex-array transducers in assessing the enhancement characteristics of suspicious breast lesions at contrast-enhanced ultrasound (CEUS)

Referee’s comments

  1. General comment

          The present paper attempts to demonstrate comparisons between different transducers at CEUS systems with purpose to assess the enhancement characteristics of suspicious breast lesions. The topic is interesting, however, to my opinion, there are several issues that degrade the submitted study and need further explanation and better presentation. The introduction section is quite poor, the text lacks information regarding the status of breast cancer, the survival rates, the golden standard techniques, their limitations, the necessity of further examination etc. Another important issue is the comparison of linear and convex-array transducers. The authors seek to assess the different performance of the aforementioned transducers on breast lesion detectability, however, there is no clear or obvious comparison presented in their results. Maybe for comparison reasons table 2, 3,4 as well as table 5,6 could be merged to one table. Then the authors should try to comment qualitatively on the numerical data provided in tables and not only present the coefficient values obtained from the readers. An effort is carried out in the discussion section but it seems that the findings just consistent with literature data. If so, then, which is the added value of the present investigation? To strengthen the validity of the present study, comparison data with other found in literature should clearly presented. In addition, to strengthen the originality of your work the new findings should clearly underlined or highlighted. Overall, the authors are called upon to significantly improve the current version of their manuscript.

Below a few specific comments are addressed.

  1. Specific comments
  2. You could include in Figure 1 the contraindications for the administration of contrast media, described in lines 68-73.
  3. The three readers were physicians or radiologists?

As my mother tongue is not English i do not propose major language modifications but i feel that English editing would be useful. A few modifications are addressed below:

  1. Line 14: Change: “…The purpose of this study is compare the…” to “…The purpose of this study is compare the…”
  2. Line 74: Change: “…we had the pathology reports, that in the case of malignant lesions…” to “…we had the pathology reports, such as the case of malignant lesions…”.
  3. Lines 165-168. Text needs rephrasing.
  4. Line 169: Change: “…with literature data, especially considering that most…” to “…with literature data. In particular, the most…”
  5. Lines 193-198. Text needs rephrasing.
  6. Line 251. Text appears suddenly in the discussion section. Text in lines 251-254 needs rephrasing or better connection with previous analysis.
  7. Try to derive a clearer conclusion from your results, especially regarding the comparison of linear and convex transducers.

Author Response

Manuscript ID: Diagnostics - 1616884

Title:  Comparison of linear and convex-array transducers in assessing the enhancement characteristics of suspicious breast lesions at contrast-enhanced ultrasound (CEUS)

Referee’s comments 

  1. General comment

    Point 1: The present paper attempts to demonstrate comparisons between different transducers at CEUS systems with purpose to assess the enhancement characteristics of suspicious breast lesions. The topic is interesting, however, to my opinion, there are several issues that degrade the submitted study and need further explanation and better presentation. The introduction section is quite poor, the text lacks information regarding the status of breast cancer, the survival rates, the golden standard techniques, their limitations, the necessity of further examination etc.

Response Point 1: Thank you for the suggestions. As a response to Point 1, we added in the body text the following statements:”

“Sonovue®has a bubble size distribution between 0.7–10 μmin diameter. Gorge et al found that microbubbles smaller than 2 μmrepresent less than 5% of the global gas volume, and their effects was negligible at frequencies up to 7 MHz. Furthermore, 80% of the echogenicity was provided by bubbles with 3-9 μmdiameter, which proved to be well adapted to clinical applications in the usual medical frequency range (1-7MHz).” (lines 78-82)

“To the best of our knowledge, the studies published in the literature regarding the assessment of breast lesions at CEUS consist in the use of linear-array transducer. Starting from the idea that the convex-array transducer, having lower frequencies could be more suitable in the evaluation of breast lesions, the originality of the present study is represented by the comparison between the two types of transducers regarding the evaluation of suspicious breast lesions.

The clinical benefit of this study would be to adapt the CEUS examination protocol of in order to obtain reliable and reproducible results.” (lines 88-95)

Regarding the status of breast cancer we mentioned the mean lesion size and the pathological types in Table 1. Do you mean we should also add the TNM classification?

The purpose of the study was not to evaluate the survival rate (considering the fact that this was a prospective study that ended in August 2021) or the necessity of further examination. All the patients were completely evaluated in the clinical practice in order to receive the appropriate treatment, but this study focused only on the CEUS examination.

We did not have a golden standard technique (none of these patients performed MRI for example in order to use it as a golden standard technique).

Point 2:  Another important issue is the comparison of linear and convex-array transducers. The authors seek to assess the different performance of the aforementioned transducers on breast lesion detectability, however, there is no clear or obvious comparison presented in their results.

Response Point 2: The purpose of the present study was only to compare the enhancement pattern of suspicious breast lesions with different type of transducers, not to evaluate the detectability and the diagnostic performance of the technique, so we did not have data regarding the detectability.

Point 3: Maybe for comparison reasons table 2, 3,4 as well as table 5,6 could be merged to one table.

Response Point 3: Thank you for the suggestion. We merged Tables 2 3 and 4 and 5 and 6 respectively.

Point 4: Then the authors should try to comment qualitatively on the numerical data provided in tables and not only present the coefficient values obtained from the readers. An effort is carried out in the discussion section but it seems that the findings just consistent with literature data. If so, then, which is the added value of the present investigation? To strengthen the validity of the present study, comparison data with other found in literature should clearly presented. In addition, to strengthen the originality of your work the new findings should clearly underlined or highlighted. Overall, the authors are called upon to significantly improve the current version of their manuscript.

Response Point 4: Thank you for the suggestions. As a response to Point 4, we added in the body text the following statements:”

“To the best of our knowledge, the studies published in the literature regarding the assessment of breast lesions at CEUS consist in the use of linear-array transducer. Starting from the idea that the convex-array transducer, having lower frequencies could be more suitable in the evaluation of breast lesions, the originality of the present study is represented by the comparison between the two types of transducers regarding the evaluation of suspicious breast lesions. The clinical benefit of this study would be to adapt the CEUS examination protocol of in order to obtain reliable and reproducible results.” (lines 88-95)

“These findings were consistent with literature data considering the evaluation of the lesions through linear-array transducer. In particular, most of the lesions included in the study (88.57%) were found to be malignant after biopsy (3,6–9). Vraka et al.(3)found that 96.3% of lesions had a inhomogeneous enhancement, 64.5% had ill defined margins, 76.5% had perilesional enhancement and all the lesions had perfusion defects and a centripetal uptake pattern when assessing them with the linear-array transducer. In our study at linear-array and convex-array transducer, according to the first, second and third reader, 98.1%, 100% and 100% of lesions were inhomogeneous, 95,5%, 96,7% and 100% had ill defined margins, 81%, 75% and 85,2% had perilesional enhancement, 98.1%, 100% and 100% had perfusion defects and 95,7%, 100% and 100% of the lesions had centripetal uptake pattern.” (lines 262-269)

Below a few specific comments are addressed.

  1. Specific comments
  2. You could include in Figure 1 the contraindications for the administration of contrast media, described in lines 68-73.

Response Point 1: Thank you for the suggestions. As a response to Point 1, we included in Figure 1 the contraindications for the administration of contrast media

  1. The three readers were physicians or radiologists?

Response Point 2: Thank you for the suggestions. As a response to Point 2 we replaced in the body text the word “physicians” with “radiologists”: “All recorded examinations were independently evaluated by three radiologists” (line 182)

As my mother tongue is not English i do not propose major language modifications but i feel that English editing would be useful. A few modifications are addressed below:

  1. POINT 3: Line 14: Change: “…The purpose of this study is compare the…” to “…The purpose of this study is compare the…”

Response Point 3: Thank you for the suggestions. As a response to Point 3, we modified in the body text the following statements: “The purpose of this study is to[IB1] compare the agreement between different types of transducers with high frequency and low frequency regarding the evaluation of the enhancement pattern at contrast-enhanced ultrasound (CEUS) of suspicious breast lesions” (lines 14-16)

  1. POINT 4: Line 74: Change: “…we had the pathology reports, that in the case of malignant lesions…” to “…we had the pathology reports, such as the case of malignant lesions…”.

Response Point 4: Thank you for the suggestions. As a response to Point 4, we modified in the body text the following statements: “For all biopsied lesions we had the pathology reports, such asthe case of malignant lesions: (lines 114)

  1. POINT 5: Lines 165-168. Text needs rephrasing.

Response Point 5: Thank you for the suggestions. As a response to Point 5, we modified in the body text the following statements: “Our results showed that most of the lesions evaluated by all three readers with both high frequency and low frequency transducers showed hyper-enhancement, non-circumscribed margins, perfusion defects, centripetal uptake pattern, and perilesional enhancement” (255-258)

  1. POINT 6: Line 169: Change: “…with literature data, especially considering that most…” to “…with literature data. In particular, the most…”

Response Point 6: Thank you for the suggestions. As a response to Point 6, we modified in the body text the following statements: “These findings were consistent with literature data. In particular, the most of the lesions included in the study” (lines 259-260)

  1. POINT 7: Lines 193-198. Text needs rephrasing.

Response Point 7: Thank you for the suggestions. As a response to Point 7, we modified in the body text the following statements: “The inhomogeneous appearance of lesions when evaluated with the linear-array transducer and the homogeneous appearance when evaluated with the convex-array transducer could be explained by the fact that microbubbles were made to resonate better at lower frequencies or the lower resolution of convex-array transducer gives the impression of homogeneous filling of the lesions” (lines 366-371)

  1. POINT 8: Line 251. Text appears suddenly in the discussion section. Text in lines 251-254 needs rephrasing or better connection with previous analysis.

Response Point 8: Thank you for the suggestions. As a response to Point 8, we modified in the body text the following statements:“Radiomics is a new field that has proven to be useful in radiology, even when applied to different imaging techniques, to distinguish between benign and malignant lesions or even between different histopathological types of cancer by using artificial intelligence.” (lines 454-457)

The idea was that if radiomics proved to be successfully used in all imaging techniques, it could be valuable applied to CEUS images.

  1. POINT 9: Try to derive a clearer conclusion from your results, especially regarding the comparison of linear and convex transducers.

Response Point 9: Thank you for the suggestions. As a response to Point 9 we modified the conclusion text the following statements“The evaluation of suspicious breast lesions with high-frequency linear-array transducer and low-frequency convex-array transducer was comparable in terms of uptake pattern and perilesional enhancement. The agreement regarding the evaluation of the degree of enhancement, the internal homogeneity, and the perfusion defects varied between fair and substantial. For all CEUS characteristics, the inter-observer agreement between all readers was superior for linear-array transducer, which leads to more a homogeneous and reproducible interpretation.” (lines 464-470)

Round 2

Reviewer 1 Report

The authors have improved the manuscript significantly and addressed my concerns reasonably well. The significance of the study is still obscure but in the revised version there is now a point that emerges and merits better/clearer description.

Specifically, the authors refer to the study by Wang et al of 2016 (lines 360-392). So, contrary to what they write in their response to my Point 1, "to the best of their knowledge" and in lines 83-84 of the revised manuscript, there is a study that compares high and low frequency transducers for benign and malignant lesions. The results from the 2016 study seem to agree with the results of this one except when readers are concerned. Readers seem to perform better with CEUS and the high frequency transducer.  It is this reader preference, which is evaluated through the inter-observer agreement, that possibly makes this study different from prior ones and, if so, this has not been clearly presented. Just reread the first and last sentence in the abstract and you will understand my point. 

"The purpose of this study is to compare the agreement between ... transducers ... regarding the evaluation ..." You studied agreement between observers and not transducers. Shouldn't this be something like: "The purpose of this study was to determine observer agreement in assessing the enhancement pattern of suspicious breast lesions with CEUS using high and low frequency transducers"? So then the statements in lines 34-37 make sense.

In addition, the originality of the present study is the observer comparison and not what is stated in lines 86-88. I also recommend that "by three experts" is added to line 417 after "... breast lesions ..." or somewhere in the first sentence of the conclusions.

Finally, still need some language editing, for example:

1) line 36, remove "between all readers". 

2) change "suspect" to "suspicious" lesions throughout the paper.

3) Lines 89-90 should be corrected ... what is "of in order"?

Author Response

The authors have improved the manuscript significantly and addressed my concerns reasonably well. The significance of the study is still obscure but in the revised version there is now a point that emerges and merits better/clearer description.

Point 1

Specifically, the authors refer to the study by Wang et al of 2016 (lines 360-392). So, contrary to what they write in their response to my Point 1, "to the best of their knowledge" and in lines 83-84 of the revised manuscript, there is a study that compares high and low frequency transducers for benign and malignant lesions. The results from the 2016 study seem to agree with the results of this one except when readers are concerned. Readers seem to perform better with CEUS and the high frequency transducer.  It is this reader preference, which is evaluated through the inter-observer agreement, that possibly makes this study different from prior ones and, if so, this has not been clearly presented. Just reread the first and last sentence in the abstract and you will understand my point. 

Response 1: Response Point 1: Thank you for the suggestions. As a response to Point 1, we modified in the body text the following statement: “To the best of our knowledge, there is only a study published in the literature regarding the assessment of breast lesions at CEUS by using the linear-array and convex-array transducer. “(lines 95-98)

Point 2:

"The purpose of this study is to compare the agreement between ... transducers ... regarding the evaluation ..." You studied agreement between observers and not transducers. Shouldn't this be something like: "The purpose of this study was to determine observer agreement in assessing the enhancement pattern of suspicious breast lesions with CEUS using high and low frequency transducers"?

So then the statements in lines 34-37 make sense.

Response Point 2: Thank you for the suggestions. As a response to Point 2, we modified in the body text the following statement:

“The purpose of this study was to determine the observer agreement in assessing the enhancement pattern of suspicious breast lesions with contrast-enhanced ultrasound (CEUS) using high and low frequency transducers. “(lines 14-16)

Point 3:

In addition, the originality of the present study is the observer comparison and not what is stated in lines 86-88.

Response Point 3: Thank you for the suggestions. As a response to Point 3, we modified in the body text the following statement:

“the originality of the present study is represented by the comparison of observer agreement regarding the evaluation of suspicious breast lesions by using the two types of transducers.” (lines 99-102)

Point 4:

I also recommend that "by three experts" is added to line 417 after "... breast lesions ..." or somewhere in the first sentence of the conclusions.

Response Point 4: Thank you for the suggestions. As a response to Point 4, we added in the abstract and in the body text the “by three expert” (lines 32 and 475)

Finally, still need some language editing, for example:

  • line 36, remove "between all readers". 

Response Point 1: Thank you for the suggestions. As a response to Point 1, we removed from the body text “between all readers” (lines 36 and 480)

  • change "suspect" to "suspicious" lesions throughout the paper.

Response Point 2: Thank you for the suggestions. As a response to Point 2, we changed in the body text the word “suspect” with “suspicious”

  • Lines 89-90 should be corrected ... what is "of in order"?

Response Point 3: Thank you for the suggestions. As a response to Point 3, we deleted “of” (line 103)

Dear reviewer, we would like to thank you for taking the time to carefully evaluate our article. Thank you for your suggestions.

Best regards,

Ioana Boca (Bene)

Reviewer 2 Report

Manuscript ID: Diagnostics - 1616884

Title:  Comparison of linear and convex-array transducers in assessing the enhancement characteristics of suspicious breast lesions at contrast-enhanced ultrasound (CEUS)

Referee’s comments

Please check the quality of table 4 before publication. Something goes wrong with the first column of the table.

Author Response

Referee’s comments 

Please check the quality of table 4 before publication. Something goes wrong with the first column of the table.

Response: Thank you for the suggestion. We modified the table.

Dear reviewer, we would like to thank you for taking the time to carefully evaluate our article. Thank you for your suggestions.

Best regards,

Ioana Boca (Bene)